# Evaluating Recycled Concrete Aggregate and Sand for Sustainable Construction Performance and Environmental Benefits

Saurabh Singh [1], Suraj Kumar Singh [2,*], Mohamed Mahgoub [3], Shahnawaz Ahmed Mir [1], Shruti Kanga [4], Sujeet Kumar [5], Pankaj Kumar [6] and Gowhar Meraj [7,*]

1   Department of Civil Engineering, Suresh Gyan Vihar University, Jaipur 302017, India; saurabh.singh@mygyanvihar.com (S.S.); shahnawaz.ahmed@mygyanvihar.com (S.A.M.)
2   Centre for Sustainable Development, Suresh Gyan Vihar University, Jaipur 302017, India
3   Red Sea Global, An Nakheel, Riyadh 12382, Saudi Arabia; momahgoub@gmail.com
4   Department of Geography, School of Environment and Earth Sciences, Central University, Punjab 151401, India; shruti.kanga@cup.edu.in
5   Department of Mechanical Engineering, Vivekananda Global University, Jaipur 302012, India; kumar.sujeet@vgu.ac.in
6   Institute for Global Environmental Strategies, Hayama 240-0115, Japan; kumar@iges.or.jp
7   Department of Ecosystem Studies, Graduate School of Agricultural and Life Sciences, The University of Tokyo, Tokyo 113-8654, Japan
*   Correspondence: suraj.kumar@mygyanvihar.com (S.K.S.); gowharmeraj@g.ecc.u-tokyo.ac.jp (G.M.)

**Abstract:** This research investigates the potential of utilizing recycled concrete aggregate (RCA) and recycled sand (RS), derived from crushed concrete cubes, as sustainable alternatives in construction materials. The study comprehensively evaluates the properties of RCA and RS, focusing on workability, impact resistance, abrasion resistance, and compressive strength to determine their viability as substitute construction materials. A notable finding is RS's enhanced fire and heat resistance when used as a fine aggregate in mortar blends, mixed with cement and Sinicon PP in a 3:1 ratio. The experimental analysis included thorough assessments of uniformity, durability, and curing time, alongside Scanning Electron Microscopy (SEM) for structural examination. Results show that RCA has an Aggregate Impact Value (AIV) of 5.76% and a Los Angeles Abrasion Value (LAA) of 21.78%, demonstrating excellent strength of the recycled aggregates. The mortar mix was also prepared using recycled sand, cement, and Sinicon PP, and its stability was confirmed through soundness tests, which resulted in a 0.53 mm expansion and a satisfactory consistency level of 44%. Ultrasonic pulse velocity (UPV) tests also indicated high-quality concrete formation using RCA and RS. SEM imaging corroborated this by revealing a bond between the cement paste and the aggregates. Incorporating RS and RCA in concrete mixtures impressively yielded a compressive strength of 26.22 N/mm$^2$ in M20-grade concrete. The study concludes that using RCA and RS waste materials in the construction sector underlines that sustainable practices can be integrated without compromising material quality. This approach aligns with sustainable development goals and fosters a more environmentally friendly construction industry.

**Keywords:** sustainable development goals; recycled sand; recycled concrete aggregate; Sinicon PP; SEM analysis; aggregate impact value; Los Angeles abrasion test





## 1. Introduction

The construction sector is crucial for global advancement but currently confronts substantial environmental challenges, mainly its significant carbon [1,2]. The ecological burden is heightened by transporting natural aggregates over long distances, escalating construction expenses, and depleting natural resources [3,4]. In order to address this, there is a pressing need for sustainable and environmentally friendly alternatives in construction

practices [5–7]. Makul et al. (2021) examine the utilization of recycled concrete aggregates (RCAs) as a sustainable substitute for natural aggregates, emphasizing its potential in high-performance concrete. The study advocates for additional research on the durability and performance improvement of RCA, highlighting its importance in decreasing construction waste and encouraging sustainable construction practices [8]. Katar et al. (2021) explored the use of recycled concrete aggregate (RCA) in self-compacting concrete (SCC), testing mixes with 0–75% RCA replacement. Findings include achievable compressive strengths over 40 MPa, decreased strength, and increased water absorption with higher RCA levels. Optimal properties were noted at 50% RCA, indicating its viability for sustainable construction materials [9]. Neupane et al. (2023)'s review emphasizes the potential of recycled concrete aggregate (RCA) and recycled aggregate concrete (RAC) in Southeast Asia, highlighting environmental benefits and challenges in structural applications. Despite their lower mechanical strengths compared to natural concrete, strategies for improvement and standardization are discussed, underscoring the need for further research and enhanced recycling practices [10]. This pursuit is in harmony with Sustainable Development Goal-9 (SDG-9), which promotes resilient infrastructure and sustainable industrialization [11,12]. Adamson et al. (2015) evaluate the environmental impact of using recycled concrete aggregates in concrete production through life cycle assessment, finding that such materials maintain acceptable environmental performance and contribute to waste reduction and land conservation. The study highlights the efficiency of terracotta concrete samples in particular, due to their low environmental impact and potential for sustainability in construction practices [13]. Roh et al. (2020) assess the environmental impacts of using recycled and by-product aggregates in concrete, finding slag and bottom ash aggregates reduce impacts compared to natural aggregates, with costs estimated between USD 5.88 to 8.79/$m^3$, highlighting their potential for sustainable construction [14]. Anjam et al. (2020) investigate the use of recycled aggregates from construction and stone factory wastes in concrete production, addressing the growing issue of waste in societies. Through laboratory tests assessing slump, compressive and tensile strength, and the modulus of elasticity, the research identifies optimal replacement percentages for these recycled materials. It concludes with a proposed mix design for recycled aggregate concrete, offering a solution to utilize waste effectively and reduce environmental impacts [15]. Serres et al. (2016) use a life cycle assessment to evaluate the environmental impact of concrete made with recycled concrete aggregates, comparing it to natural aggregates. It finds that recycled materials, including concrete and terracotta brick aggregates, exhibit lower environmental impacts due to factors like reduced transport needs and enhanced durability. This underscores the benefits of using recycled materials in construction for reducing waste and conserving resources [16]. Restuccia et al. (2016) investigated using recycled sand (RS) from construction and demolition waste in mortar, finding that washing and sieving RS improves its quality and the mortar's mechanical properties. This innovative approach enhances sustainability in construction by effectively reusing waste, thereby aligning with environmental conservation efforts [17]. Łukowski et al.'s 2024 research demonstrates that replacing natural sand with recycled sand in cement mortar offers significant environmental and economic benefits without notably affecting its technical properties, thereby promoting sustainability in construction [18]. Conventional concrete, when utilizing brick or recycled brick aggregates, typically exhibits lower mechanical strength and reduced durability than stone aggregates [19,20]. This is attributed to bricks' higher porosity and decreased strength and density [21,22]. In alignment with SDG-11, which aims for sustainable cities and communities, research has investigated the use of induction furnace slag (IFS) as an alternative to brick aggregates in concrete [23,24]. This initiative contributes to developing inclusive, safe, resilient, and sustainable urban environments. Studies indicate that IFS' higher density and lower water absorption rate improve concrete properties, furthering sustainable construction methods [25]. Such an approach also advances SDG-12, which calls for responsible consumption and production by enhancing resource efficiency and reducing waste [26]. Incorporating IFS into concrete has been shown to bolster mechani-

cal strength and durability, marking it as a viable option for environmentally conscious construction efforts. These developments also align with SDG-13, which focuses on taking urgent action to combat climate change (ESCAP 2019).

The literature on environmentally sustainable construction practices is rich with strategies for enhancing concrete properties [27]. One notable approach is employing hemp fiber rope (HFR) as a reinforcement in recycled brick aggregate concrete (RAC-FCSB), which has been shown to improve the compressive stress–strain behavior substantially [28]. Research identifies two effective HFR reinforcement techniques: strip reinforcement and complete wrapping. Both methods have been shown to reduce the construction industry's carbon footprint, a vital goal in combating climate change. Complementary research extends into the evaluation of warm mix asphalt incorporating recycled concrete (WMA-RCA), suggesting it as a sustainable alternative in pavement engineering. The utilization of glass waste in concrete not only diverts it from landfills but also contributes to the development of novel structural materials [29]. Furthermore, sustainable mortar formulations incorporating waste elements have been thoroughly investigated, suggesting significant ecological and structural benefits [30–32]. A broader literature review confirms the increasing acceptance of recycled aggregates in various concrete applications, with evidence supporting their use in standard and advanced concrete systems [33]. The inclusion of supplementary cementitious materials is acknowledged for their role in enhancing the durability and strength of concrete, thus contributing to more sustainable construction practices.

Recent studies have highlighted geopolymers and alkali-activated materials as environmentally friendlier substitutes for conventional Portland cement concrete, maintaining structural integrity with a lower environmental footprint [34]. These advancements represent a paradigm shift in sustainable construction practices. Employing waste materials and industrial by-products has emerged as a critical approach to mitigate the environmental effects of concrete production [34,35]. Moreover, technological progress is evident in the surface modification of recycled aggregates, like waste glass, which is proven to improve concrete performance substantially [36]. The existing body of research is well aligned with the SDGs, particularly those related to sustainable urban development, responsible resource usage, and climate mitigation efforts. This suggests a unified movement towards embracing advanced materials and methods that enhance construction practices' environmental and structural integrity [37]. However, a significant knowledge gap persists regarding the properties and potential of cement mortar made with recycled sand (RS) derived from crushed concrete. The complex interactions between RS and RCA within concrete mixtures have not been thoroughly investigated [38,39]. Additionally, the influence and effectiveness of Sinicon PP, a unique volcanic glass found at only one location on Earth in a large deposit, which is South Africa, needs further investigation. Sinicon Sand is made out of feed from these mines using a manufacturing process to convert this volcanic glass into well-sealed tough glass granules, which is ideally suited for use with cementitious and other binders [40] This was used in the preparation of the mortar with a ratio of 1:3. The novelty of this study lies in the investigation of properties of recycled concrete aggregate (RCA) and recycled sand (RS) derived from freshly prepared and crushed concrete cubes, contrasting with previous studies that sourced RCA from much older demolished structures (approximately 40–70 years old). This fresh perspective on the age of materials provides insights into the early-age properties of RCA and RS, combined with the innovative use of Sinicon PP in mortar mix preparation, offering a new dimension to the understanding and application of these materials in sustainable construction practices. This study aims to rigorously examine the properties of cement mortar and concrete incorporating RS, RCA, and Sinicon PP. Through comprehensive testing, the research intends to uncover the detailed characteristics of these materials, focusing on their structural resilience and mechanical properties.

## 2. Materials and Methods

For this study, the primary materials of interest were the RCA and RS sourced from M20- and M30-grade crushed concrete cubes obtained from the construction sites of

Acharya Shri Purushottam Uttam Medical College and Hospital. In this study, M20-grade concrete was used in a 1:1.5:3 mix ratio, meaning 1 part cement, 1.5 parts RS, and 3 parts RCA. Table 1 presents the M20 mix designs in accordance with IS-10262-2009 [41]. The obtained concrete aggregates were freshly tested at the age of 28 days, and these crushed concrete cubes were collected from the construction site between 35 and 40 days after casting. These concrete cubes were then further crushed into smaller particles using a Concrete Waste Crushing Machine (12.5 HP). Following the crushing process, the resultant material underwent a sieving procedure. This step was crucial for classifying the aggregates by size, thereby facilitating the assessment of their suitability for various applications based on granularity. Figure 1 shows coarse aggregates sieved in sizes of 20 mm, 10 mm, and 5 mm. In contrast, fine aggregates were defined by particles passing through a 4.75 mm sieve, retaining those finer in composition. The crushed concrete cubes served as the primary source of RCA. RS possesses several critical physical properties that make it a valuable material in construction and engineering applications, as shown in Table 1. With a low water absorption rate of 6.2% and a moisture content of just 0.5%, it resists the detrimental effects of excess moisture, ensuring stability and durability. Its bulk density, measuring 1740 kg/m$^3$, signifies a compact and dense composition, providing essential support in construction.

**Table 1.** M20 Concrete Mix Design Parameters as per IS-10262-2009 Standards.

| Grade Designation | M20 |
|---|---|
| Target Strength | 20 N/mm$^2$ |
| Type of Cement | OPC 53 grade conforming to IS-12269-1987 [42] |
| Maximum Nominal Aggregate Size (RCA) | 20 mm |
| Minimum Cement Content (MORT&H 1700-3 A) | 250 kg/m$^3$ |
| Maximum Water–Cement Ratio (MORT&H 1700-3 A) | 0.5 |
| Workability (MORT&H 1700-4) | 25 mm (Slump) |
| Exposure Condition | Normal |
| Degree of Supervision | Good |
| Type of Aggregate | Crushed Angular Aggregate |
| Maximum Cement Content (MORT&H Cl. 1703.2) | 540 kg/m$^3$ |

Moreover, a specific gravity of 2.8 indicates its relative density compared to water [36]. These combined physical characteristics make RS a good choice for use in mortar, as shown in Table 2. A systematic sieving process was employed using sieves of distinct sizes to categorize the material into coarse and fine aggregates. A comprehensive breakdown of the methodology, illustrated in Figure 2, presents a detailed flowchart outlining the step-by-step process.

**Table 2.** Physical Characteristics of RS Used in Construction Applications.

| Physical Property of RS | Value |
|---|---|
| Water Absorption (%) | 6.2 |
| Bulk Density (kg/m$^3$) | 1740 |
| Specific Gravity | 2.8 |
| Moisture Content (%) | 0.5 |

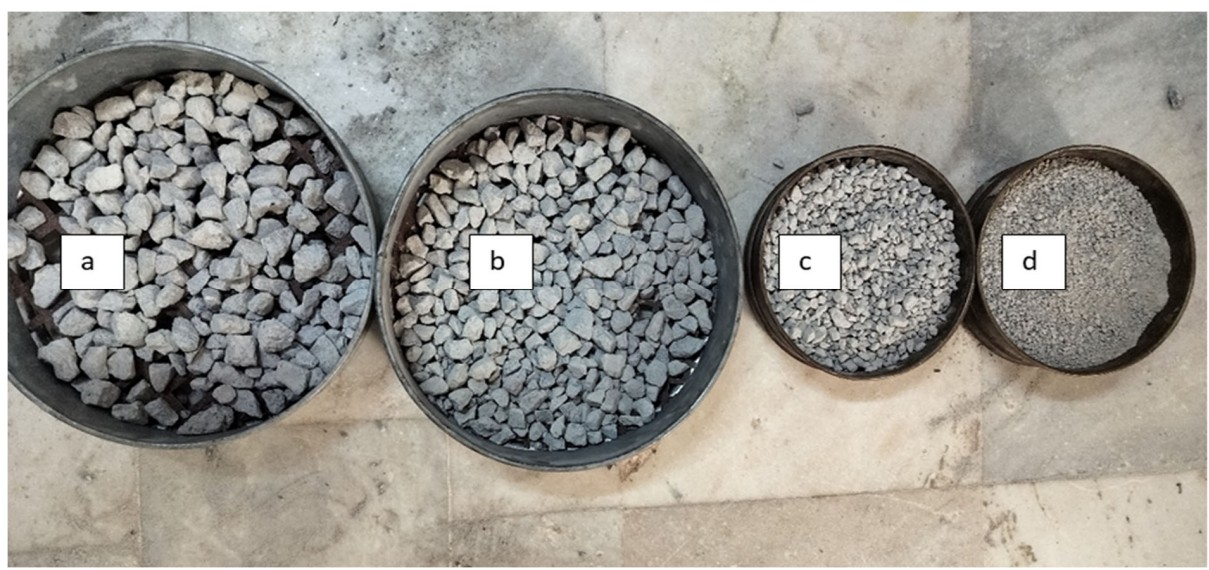

**Figure 1.** RS and aggregate classified based on their size: (**a**) 20 mm size of aggregate; (**b**) 10 mm size of aggregate; (**c**) 5 mm size of aggregate; and (**d**) >4.75 mm size of aggregate.

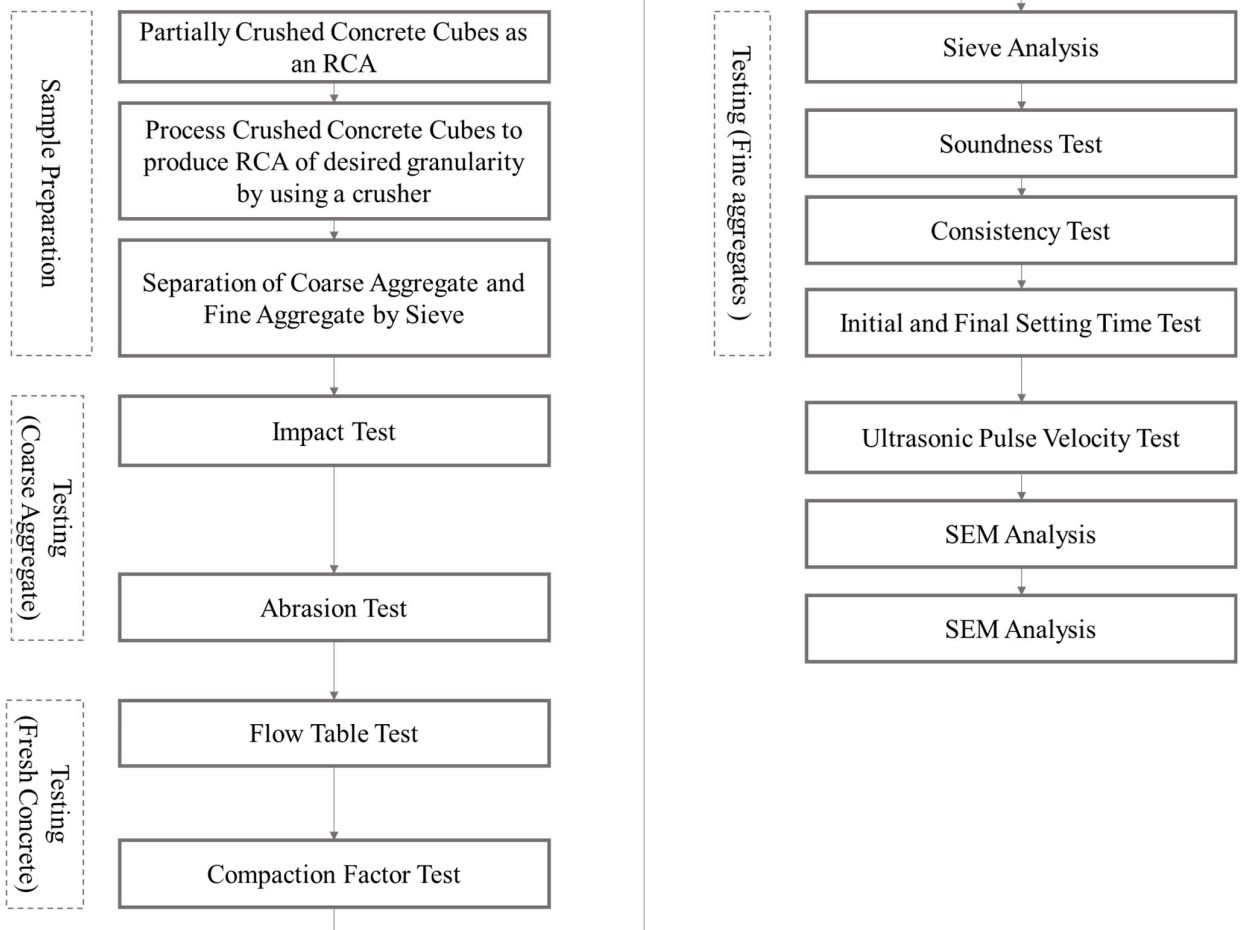

**Figure 2.** Schematic representation of the research methodology employed for evaluating RS and RCA.

*2.1. Coarse Aggregate Testing*

2.1.1. Aggregate Impact Value (AIV)

The Aggregate Impact Value (AIV) evaluates an aggregate's resilience to sudden shocks or impacts [43]. This measurement is vital in providing a relative indication of an aggregate's resistance capacity. The apparatus has an integrated counter that precisely and automatically records the number of blows administered to the aggregate sample under examination. This methodology ensures an accurate and repeatable assessment of the AIV, which is essential for determining the material's suitability for various construction applications. The AIV is mathematically represented as shown in Equation (1):

$$\text{AIV} = \frac{\text{Weight of Fines}}{\text{Original Weight}} \times 100 \tag{1}$$

In this context, "weight of fines" refers to the mass lost by the aggregate sample after undergoing the impact test. This loss is critical for calculating the Aggregate Impact Value (AIV) percentage, which quantitatively measures the material's resistance to impact forces.

2.1.2. Abrasion Test

The abrasion test is essential to assess the durability of recycled aggregate, quantifying its resistance to wear through the Los Angeles Abrasion Value (LAA) [44]. By comparing the sample's weight before and after testing, the resulting weight difference offers insight into the material's ability to resist abrasion, which is critical for evaluating its longevity and appropriateness for construction uses. The abrasion test is mathematically represented as follows:

$$\text{LAA}(\%) = \frac{\text{Weight of sample before test} - \text{Weight of sample after test}}{\text{Weight of sample before test}} \times 100 \tag{2}$$

2.1.3. Compaction Factor Test

This test is pivotal for assessing the workability of concrete mixtures [45]. It gauges the ease of compaction, determining how readily concrete can achieve its final form in construction applications. The compaction test is mathematically represented as follows:

$$\text{Compaction Factor Test} = \frac{\text{Weight of partially compacted concrete}}{\text{Weight of fully compacted concrete}} \tag{3}$$

Concrete workability refers to the ease with which fresh concrete can be mixed, placed, finished, and consolidated uniformly [45,46]. The workability of concrete depends on its rheological properties, determined mainly by the water-to-cement ratio [47]. A higher fluidity improves workability, making compaction and finishing more accessible and ensuring concrete homogeneity and integrity.

*2.2. Fine Aggregate Testing*

2.2.1. Sieve Analysis

The sieve analysis process is employed to ascertain the gradation of aggregate particles within a given sample [48], thereby assessing its conformity with design specifications, production control requirements, and verification specifications.

$$\text{Percentage finer} = \frac{\text{Cumulative weight of aggregates retained up to a sieve}}{\text{Total weight of the sample}} \times 100 \tag{4}$$

2.2.2. Soundness Test

The soundness test for mortar, a composite material of sand and cement, is determined using the Le Chatelier apparatus. The assessment of the potential for expansion or

volume changes in the mortar mixture is of utmost importance, making this test highly significant [49].

$$\text{Expansion} = \text{Final Reading} \ (l_1) - \text{Initial reading} \ (l_2) \tag{5}$$

This measurement of expansion provides valuable information about the stability and durability of the mortar mixture, helping to ensure its performance and reliability in construction applications.

### 2.2.3. Consistency Test

The consistency test assesses the percentage of water content in the mortar mixture [50]. The consistency refers to the specific water content required for the Vicat plunger to penetrate a depth ranging from 5 mm to 7 mm measured from the bottom of the Vicat mold. The test is essential in comprehending the optimal water-to-cement ratio necessary to attain the intended viscosity for diverse construction purposes. The ease of workability and maintenance of structural integrity are imperative factors in construction material testing, as they contribute significantly to the ability of cement paste or mortar to be effectively manipulated.

### 2.2.4. Setting Time Tests

Setting time tests is crucial for cementitious materials like mortar and cement paste [51]. These tests reveal how long the material takes to harden from being plastic to set. These tests have two essential measurements. The initial setting time is the time it takes the Vicat plunger to penetrate the test block 33 mm deep. When a material loses plasticity and sets, the final setting time occurs when the Vicat needle leaves an impression on the test block but does not penetrate further. This means the material has hardened and set. Setting time measurements ensure construction materials are set within a predictable and controllable timeframe, enabling efficient construction processes and meetings.

### 2.2.5. Flow Table Test

The flow test is conducted to assess the workability of concrete. The concrete's workability is assessed by evaluating its flowing property or fluidity. The flow test is a basic laboratory procedure. The functioning of this test is based on the perturbation of the standard mass of concrete, which is subsequently quantified by assessing the concrete flow. The flowability of concrete is indicative of its workability:

$$\text{Mortar Flow(F)} = \frac{D_f - D_i}{D_i} \tag{6}$$

F = Flow of Mortar (in percentage);
$D_f$ = Final flow traces of Mortar samples (in inches);
$D_i$ = Initial flow traces of Mortar sample (usually the diameter of the mold, in inches).

### 2.2.6. Ultrasonic Pulse Velocity Test

The Ultrasonic Pulse Velocity (UPV) testing method assesses the structural integrity and quality of concrete or stone elements [52,53]. This technique involves the measurement of the speed and attenuation of an ultrasonic wave as it traverses the material under examination.

$$V = \frac{D}{T} \tag{7}$$

This formula uses 'D' for the distance between ultrasonic transducers and 'T' for the pulse's travel time. Velocity measurements indicate concrete density and homogeneity; deviations from the expected velocity can point to structural flaws. This test dramatically impacts construction quality control and assurance.

2.2.7. SEM Analysis

SEM analysis reveals the aggregate microstructure in detail and qualitatively [54]. SEM uses high-resolution imaging and electron beams to study the aggregate's surface and internal composition [55]. This analysis reveals aggregate texture, porosity, mineral composition, and defects or anomalies. SEM is essential in materials science and construction quality assessment to understand fine details affecting aggregate performance and durability in various construction applications [52,56].

2.2.8. Compressive Strength of the Concrete

Determining the target strength in concrete mix design is essential to ensure that the concrete meets the necessary performance criteria for its intended use, particularly for structural elements where strength is vital. The target strength is typically established above the characteristic strength to provide a safety margin and accommodate variations in material properties and environmental factors. The target strength ($f't$) can be calculated using the formula:

$$f't = f'c + 1.65 \times \sigma \tag{8}$$

$f't$ is the target strength;
$f'c$ is the characteristic compressive strength;
$\sigma$ is the standard deviation of the concrete strength;
1.65 is a factor providing a one-sided tolerance that accounts for 95% probability (corresponding to the 5% defect rate).

**3. Results**

*3.1. AIV of Recycled Aggregates*

The AIV of the recycled aggregate was 5.76% after being calculated and determined. The final result is calculated by taking the average of the three different test samples. This result is arrived at by using the following formula:

$$\text{AIV}(\%) = \frac{34}{590} \times 100 \approx 5.76\% \tag{9}$$

The AIV metric is fundamental to determining how recycled aggregates can handle impact load. In evaluation criteria, an AIV result below the 10% mark is significant because it means that the whole is "exceptionally strong", according to the classification shown in Table 3 of the BS 812-110:1990 standard [57]. This classification implies that the recycled aggregate exhibits exceptional resilience against sudden shocks or impacts, making it exceptionally well suited for applications that require the utmost strength and durability. This finding underscores the suitability of recycled aggregate for demanding construction scenarios where robust and long-lasting materials are essential for successful project outcomes. It aligns with the criteria that place such aggregates in the "exceptionally strong" category, demonstrating their outstanding performance potential in high-stress environments.

**Table 3.** Classification and suitability criteria based on AIV for construction applications.

| AIV (%) | Suitability for Use |
| --- | --- |
| Less than 10% | Exceptionally strong |
| 10–20% | Strong |
| 20–30% | Satisfactory for road surfacing |
| More than 30% | Weak for road surfacing |

*3.2. Compaction Factor Test*

Upon conducting the compaction factor test for the sample, it was determined that the compaction factor was approximately 0.911. This value was derived from the following formula:

$$Compaction\ Factor = \frac{w3 - w1}{w2 - w1} \tag{10}$$

Equation (11) completes the picture given the following values:

w1 = 4.616 kg;
w2 = 15.800 kg;
w3 = 14.800 kg.

$$Compaction\ Factor = \frac{14.800 - 4.616}{15.800 - 4.616} \approx 0.911 \tag{11}$$

An observed compaction factor value of 0.911 falls within the typical range for compaction factors, indicating that the material is within standard parameters for workability. Typically, compaction factors range from 0.7 to 1.0, and a value of 0.911 suggests that the material is appropriately workable for its intended use. The compaction factor values are associated with different degrees of workability and typical uses in construction, as shown in Table 4 (American Society for Testing and Materials 2007).

**Table 4.** Compaction Factor Values and Their Corresponding Workability Levels with Typical Construction Applications.

| Compaction Factor | Degree of Workability | Typical Use |
|:---:|:---:|:---:|
| 0.78–0.85 | Very Low | Roads (using vibratory rollers) |
| 0.85–0.92 | Low | Lightly reinforced sections |
| 0.92–0.95 | Medium | Heavily reinforced sections with vibration |
| 0.95–1.00 | High | Thin sections, slip formwork |

*3.3. Abrasion Test*

This LAA value of approximately 21.78% signifies that the aggregate has experienced a weight reduction of about 21.78% due to abrasion and impact during the test.

$$LAA(\%) = \frac{5000 - 3911}{5000} \times 100 = 21.78\% \tag{12}$$

With an LAA value of approximately 21.78%, as indicated in Figure 3, the aggregate attains an "excellent" classification. This classification underscores the aggregate's remarkable wear resistance, rendering it a prime candidate for applications demanding top-tier quality, such as high-quality concrete formulations and heavily used pavements, including airfield runways. The exceptional durability exhibited by this aggregate, as highlighted in Table 5 (ASTM 2006), further underscores its appropriateness for construction projects where the paramount requirement is a steadfast ability to withstand wear and abrasion, ensuring sustained performance and structural integrity over the long term.

**Table 5.** The LAA value classifications and their applicability in construction.

| LAA Value (%) | Classification | Typical Use |
|:---:|:---:|:---:|
| Less than 30% | Excellent | High-quality concrete, pavements, airfield runways |
| 30–35% | Good | Ordinary concrete pavements with moderate traffic |
| 35–40% | Satisfactory | Low-strength concrete, pavements with light traffic |
| Above 40% | Doubtful | Generally considered unsuitable for most applications |

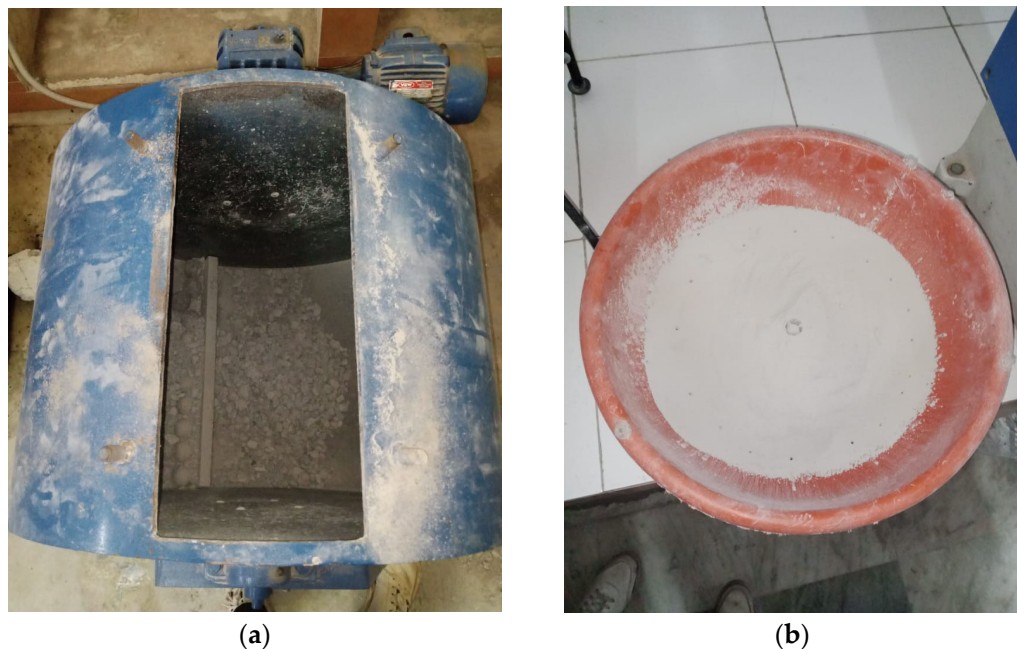

(**a**) (**b**)

**Figure 3.** Illustration of the LAA test procedure with (**a**) initial recycled aggregate setup and (**b**) post-test sieved sample.

### 3.4. Sieve Analysis

The graphical representation in Figure 4 effectively illustrates the particle size distribution of RS, comparing it to sand from all four zones. This comparison reveals a variation between Zone I and Zone II. This observation suggests that the recycled sand (RS) is suitable for masonry wall construction applications such as mortar and plastering, as indicated by the results of the sieve analysis presented in Table 6.

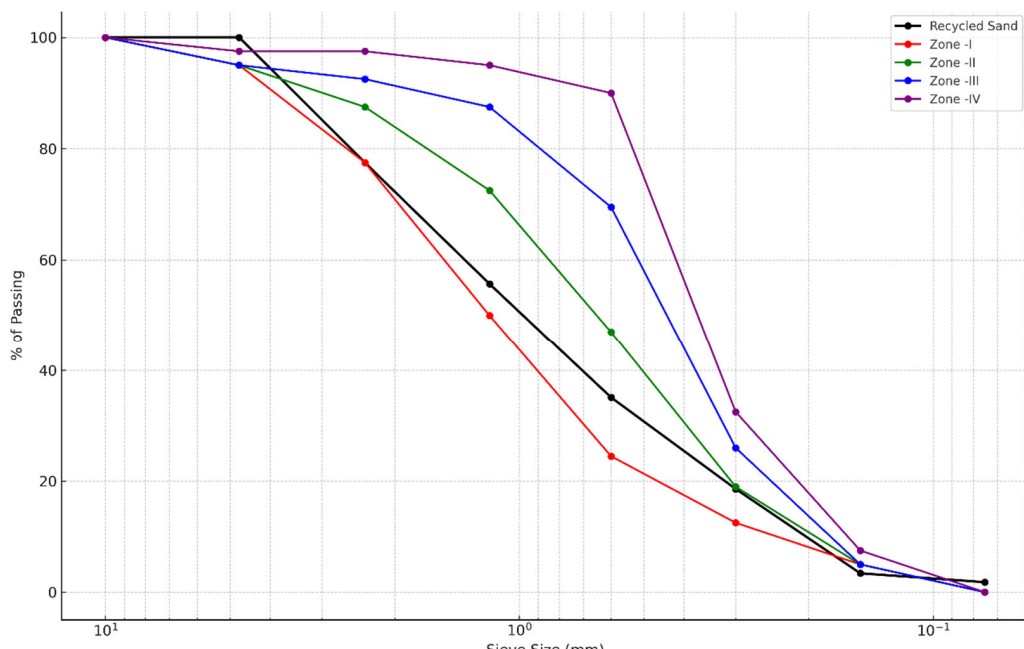

**Figure 4.** Comparative Graphical Representation of Particle Size Variations in RS vis-à-vis Traditional Recycled Sand.

**Table 6.** Particle Size Distribution Analysis of Sand Particles by Size.

| I.S. Sieve Size (mm) | Retained Weight | % of Weight Retained | Cumulative % of Weight Retained | Cumulative % of Passing |
|---|---|---|---|---|
| 10 | 0 | 0 | 0 | 100 |
| 4.75 | 0 | 0 | 0 | 100 |
| 2.36 | 225 | 22.5 | 22.5 | 77.5 |
| 1.18 | 218 | 21.8 | 44.3 | 55.7 |
| 0.6 | 206 | 20.6 | 64.9 | 35.1 |
| 0.3 | 165 | 16.5 | 81.4 | 18.6 |
| 0.15 | 152 | 15.2 | 96.6 | 3.4 |
| 0.075 | 16 | 1.6 | 98.2 | 1.8 |
| Pan | 18 | 1.8 | 100 | 0 |
| | 1000 | 100 | 3.097 | |

This comprehensive analysis of particle size distribution aids in understanding the suitability of the sand for specific construction applications, allowing for informed decisions regarding its use in various masonry and plastering projects.

*3.5. Soundness Test*

The outcome of the soundness test reveals essential insights about the aggregate's durability and capacity to withstand volume changes resulting from temperature variations. A smaller value, closer to zero, indicates a sound aggregate with robust durability characteristics. Such aggregates are less prone to significant volume fluctuations due to environmental conditions, making them well suited for construction applications, as depicted in Figure 5. The test results for three specimens are detailed in Table 7.

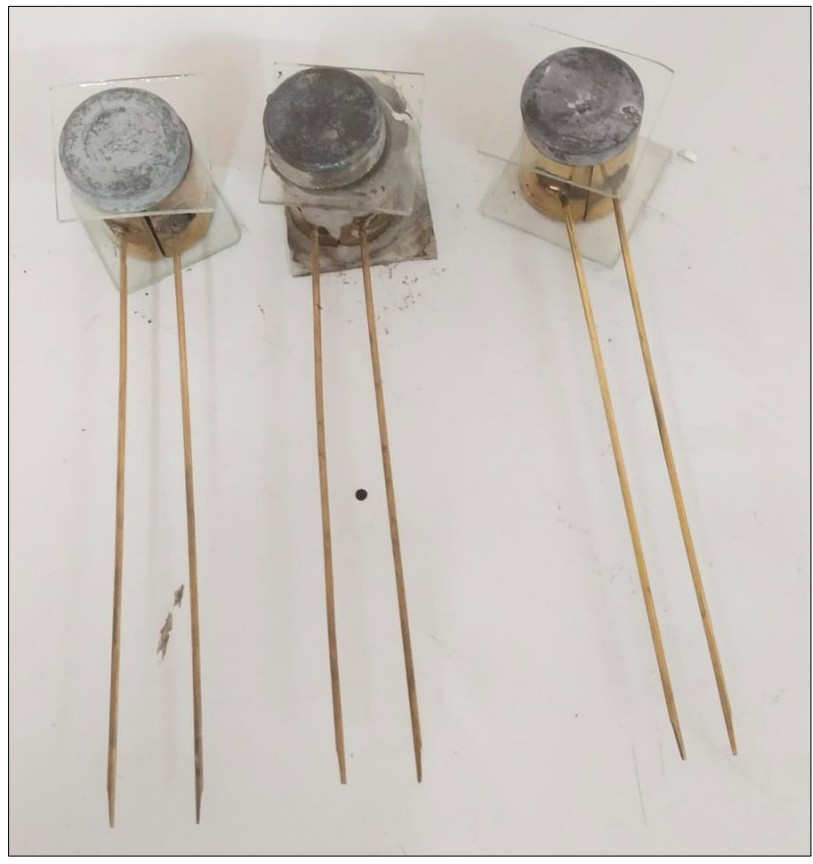

**Figure 5.** Soundness test—aggregate expansion under different temperature conditions.

The measured average expansion of 0.53 mm reflects the change in size (either expansion or shrinkage) of the aggregate when it is subjected to different temperature conditions, as specified in Table 5. During our experiments, the aggregate displayed a minimal change in size, with a difference of 0.53 mm between the two temperature extremes. This negligible change strongly indicates the recycled sand's soundness properties. It implies that the aggregate is stable and can withstand environmental variations without losing its shape or strength, thus confirming its suitability for construction projects.

**Table 7.** Results Highlighting the Stability and Expansion Properties from the Soundness Assessment of RS.

| Specimen No. | Initial Reading ($l_2$) | Final Reading ($l_1$) | Expansion = ($l_1 - l_2$) |
|---|---|---|---|
| 1 | 2.3 | 2.7 | 0.4 |
| 2 | 2.1 | 2.9 | 0.8 |
| 3 | 2.4 | 2.8 | 0.4 |
| Average | | | 0.53 |

*3.6. Consistency Test*

The outcomes of the consistency test indicate that the mortar mix achieves a consistency level of around 44% as shown in the Figure 6. This range suggests that the mortar has the right level of workability, making it particularly appropriate for use in masonry work. It is crucial to sustain this degree of consistency to ensure that the mortar maintains the perfect equilibrium between being too rigid and too fluid, which is necessary for practical application. It is neither excessively stiff, making it challenging to spread and work with, nor overly fluid, potentially compromising its strength. This consistency range is conducive to achieving a secure bond between bricks or blocks, thereby enhancing structural integrity in masonry construction. The test results from various trials are summarized in Table 8.

**Table 8.** Assessment outcomes indicate the mortar mixture's workability and consistency parameters.

| Trial No. | Percentage by Water of Dry Cement (%) | Amount of Water Added (mL) | Penetration (mm) |
|---|---|---|---|
| 1 | 13.75 | 55 | 5 |
| 2 | 16.25 | 65 | 7 |
| 3 | 18.75 | 75 | 9 |
| 4 | 21.25 | 85 | 10 |
| 5 | 23.75 | 95 | 11 |
| 6 | 26.25 | 105 | 13 |
| 7 | 28.75 | 115 | 15 |
| 8 | 30 | 120 | 18 |
| 9 | 31.25 | 125 | 20 |
| 10 | 32.5 | 130 | 22 |
| 11 | 35 | 140 | 23 |
| 12 | 36.25 | 145 | 25 |
| 13 | 37.5 | 150 | 26 |
| 14 | 38.75 | 155 | 29 |
| 15 | 40 | 160 | 32 |
| 16 | 41.25 | 165 | 34 |
| 17 | 53.75 | 215 | 36 |
| 18 | 56.25 | 225 | 39 |

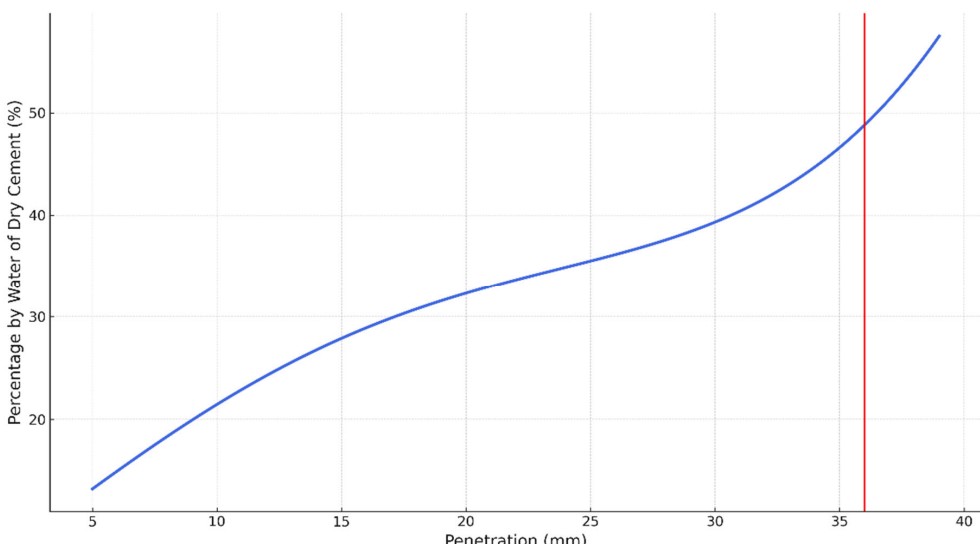

**Figure 6.** Graphical illustration depicting the standard consistency attributes of a blend comprising cement and recycled aggregate. The blue line displays the range of water content percentages and their corresponding penetration depths, indicating the optimal mixture for achieving desired workability. The red line depicts the maximum consistency attainable.

### 3.7. Initial and Final Setting Time Tests

The setting times of cementitious materials are pivotal in determining their workability, offering a clear window for various construction activities. The initial setting time, restricted by the penetration depth of 33 mm by the Vicat plunger, signifies the beginning of the loss of plasticity. This transition marks the cessation of the material's prime workability phase. For the specific mortar blend comprising cement and RS, the initial setting time was 40 min, as shown in Table 9.

On the other hand, the final setting time is ascertained when the Vicat needle leaves a mere surface impression without any further penetration into the sample. This point marks the total hardening of the material. In the studied mortar blend context, a final setting time of 630 min was observed, as visually represented in Figure 7.

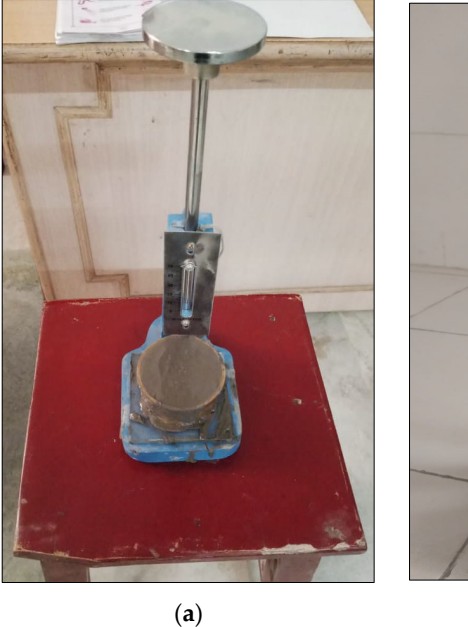
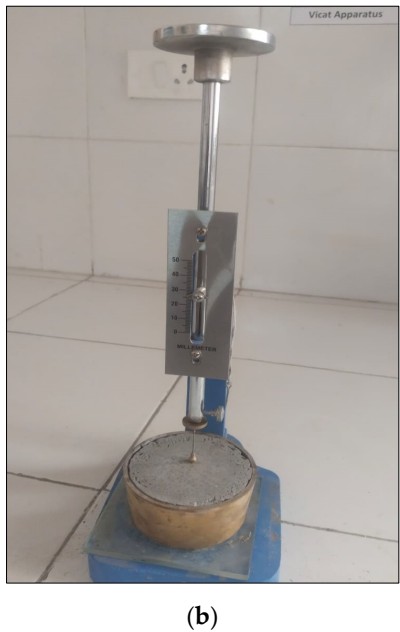

(**a**)  (**b**)

**Figure 7.** Depiction of setting times for the mortar blend (**a**) initial setting time and (**b**) final setting time.

**Table 9.** Determination of initial and final setting durations for the mortar mix incorporating cement and RS.

| Parameter | Description | Measurement/ Value |
|---|---|---|
| Initial Setting Time | Time at which Vicat plunger penetrates to a depth of 33 mm. Indicates the onset of loss of plasticity. | 40 min |
| Final Setting Time | Time at which Vicat needle leaves a surface impression without further penetration. Indicates complete hardening. | 630 min |

*3.8. Flow Table Test*

The flow table test was conducted on cement RS mortar samples to evaluate their workability, specifically in terms of flow percentage, as shown in Figure 8 and Comparative analysis of flow table test variations based on water content in RS, cement mortar, and the inclusion of Sinicon PP is shown in Figure 9. The flow of each sample was calculated using the formula shown in Table 10.

$$\text{Average Flow}(F) = \frac{30 + 25 + 27.5}{3} = 27.5\% \tag{13}$$

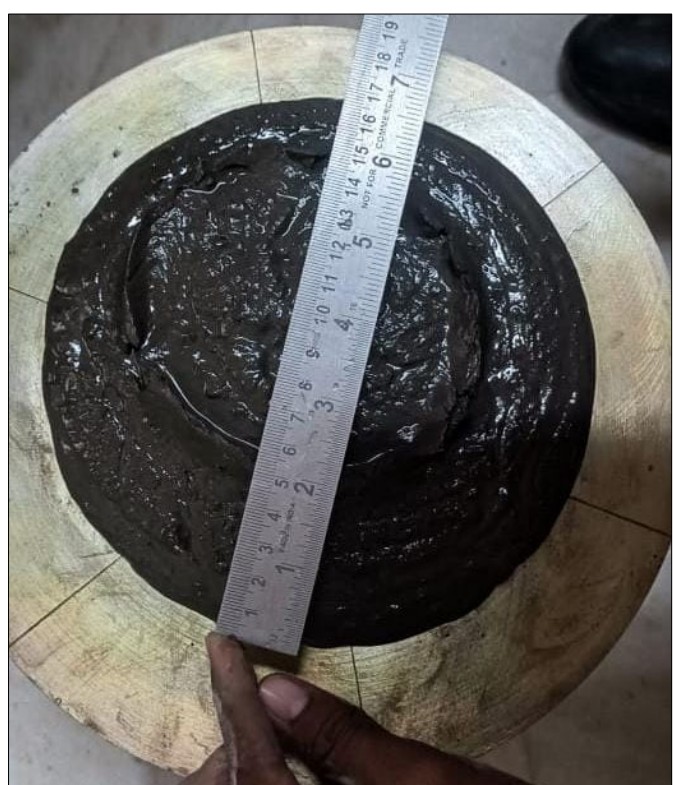

**Figure 8.** Illustration of the flow table test conducted on mortar.

**Table 10.** Workability assessment of cement–sand mortar samples derived from flow table analysis.

| Sample Number | Initial Diameter $D_i$ (in) | Final Diameter $D_f$ (in) | Flow (%) |
|---|---|---|---|
| 1 | 4.00 | 5.20 | 30.0 |
| 2 | 4.00 | 5.00 | 25.0 |
| 3 | 4.00 | 5.10 | 27.5 |

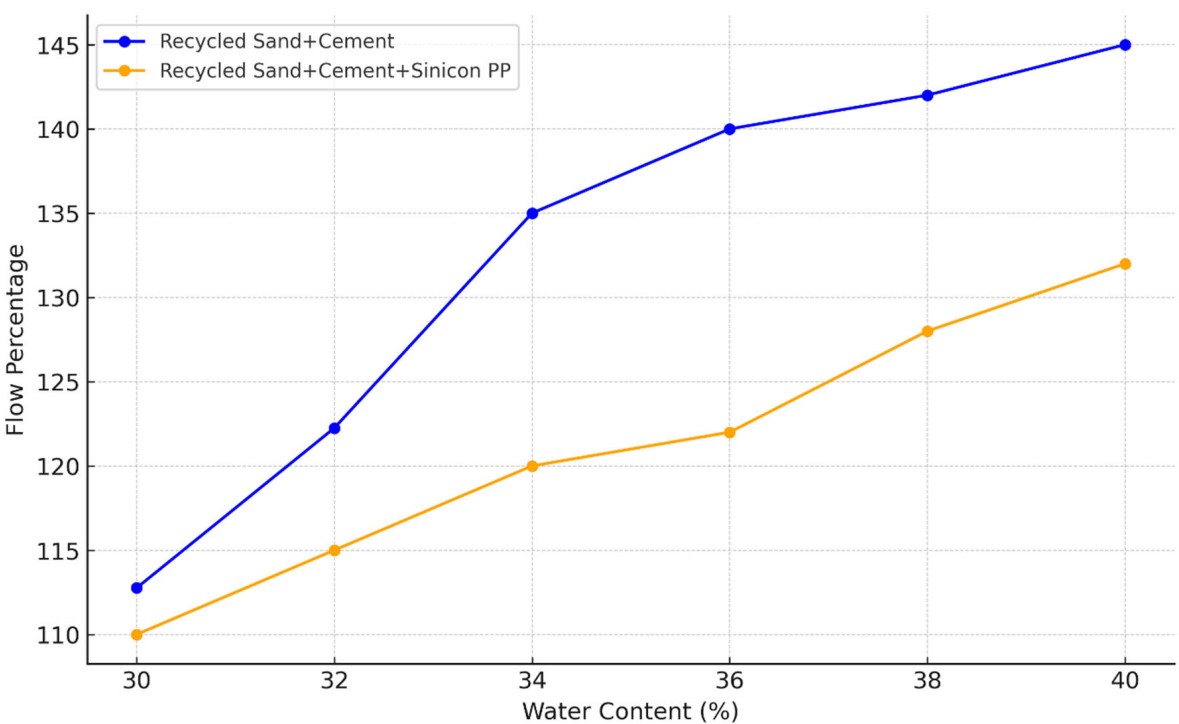

**Figure 9.** Comparative analysis of flow table test variations based on water content in RS, cement mortar, and the inclusion of Sinicon PP.

### 3.9. Ultrasonic Pulse Velocity (UPV) Test

The UPV test is a non-destructive method used to assess the quality of concrete and its components. The speed of ultrasonic pulses traveling through the concrete directly correlates with the material's density, uniformity, and quality. The test results are often used to conclude the concrete's overall integrity and potential durability. For the mix prepared using cement, RS, and RCA in a ratio of 1:1.5:3, the UPV test yielded the following results in Table 11:

**Table 11.** Evaluation Outcomes from the Ultrasonic Pulse Velocity Assessment of the Concrete Samples.

| Mix | Specimen | Weight (gm) | Density (gm/cc) | Transit Time (μs) | Pulse Speed | Concrete Quality Grading |
|---|---|---|---|---|---|---|
| Cement + RS + RCA (1:1.5:3) | A | 8690 | 2.575 | 30 | 5.00 | |
| | B | 8760 | 2.606 | 32.5 | 4.62 | Excellent |
| | C | 8000 | 2.572 | 33 | 4.55 | |
| | Average | 8722 | 2.584 | 31.83 | 4.72 | |

### 3.10. SEM Analysis of Concrete with RS

The SEM analysis conducted on concrete samples containing RS revealed significant findings, as depicted in Figure 10A. The images unveiled irregularly shaped and sized particles of recycled sand, exhibiting a strong bond with the cement paste. This bond, distinct from that of natural aggregates, indicated an enhanced level of adhesion between the RS particles and the cementitious matrix. Furthermore, the presence of well-defined edges in the particles suggested improved mechanical resistance, potentially contributing to the overall strength of the concrete.

In Figure 10B, the SEM images showcased irregularities in the RCA alongside well-defined edges. These characteristics hinted at a substantial interlocking effect between the RCA particles and the cementitious matrix, which could enhance the concrete's mechanical properties. Moreover, the uniform embedding of RS particles within the cement paste,

as observed in the SEM images, suggested that the recycled material could effectively maintain the structural integrity and durability of the concrete.

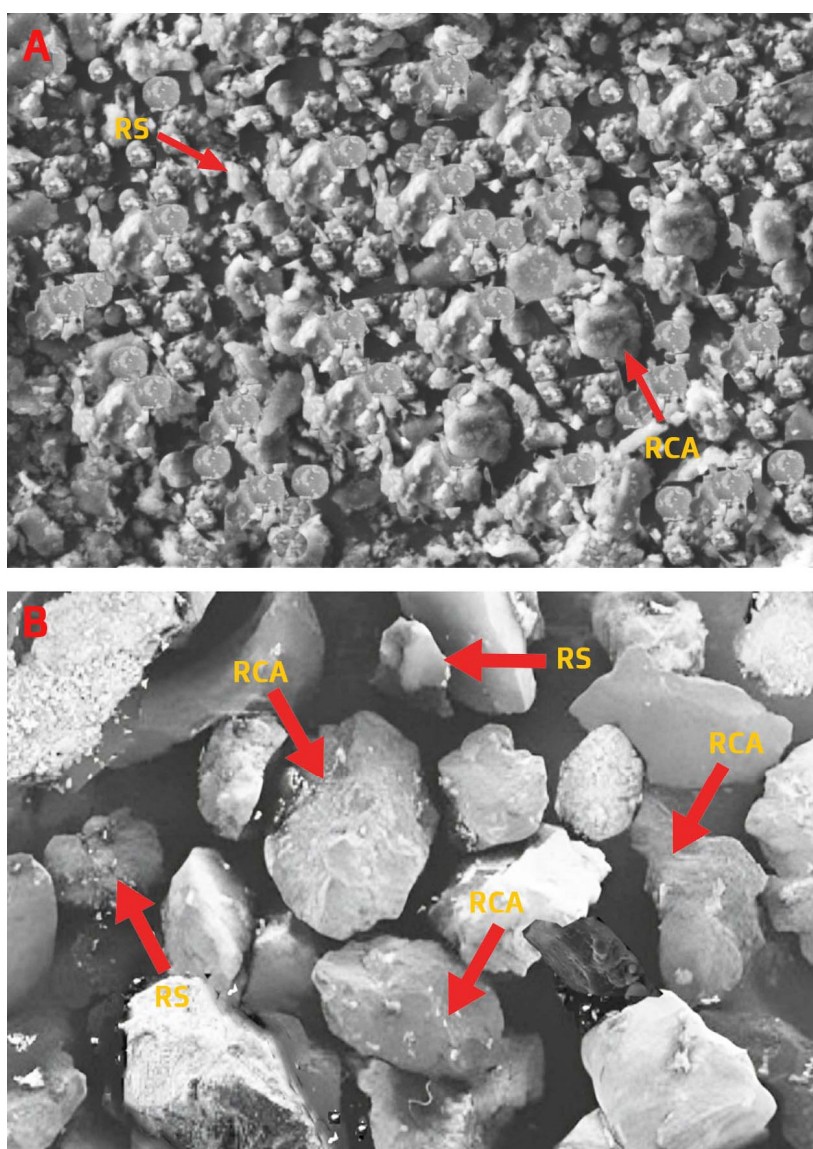

**Figure 10.** SEM examination of concrete incorporating RS and RCA (**A**) Shown at lower magnification; (**B**) Shown at higher magnification.

The concrete samples' SEM micrographs highlighted the RS microstructural attributes obtained from crushed cubes. At the Paste–RS Interfacial Transition Zone (ITZ), a prominent and cohesive bond between the cement paste and the RS was evident. With its well-embedded nature, this interface suggests an optimal integration of RS within the matrix, potentially leading to improved mechanical robustness. A closer inspection revealed that the cement paste enveloped the RS particles uniformly, signifying the material's potential to maintain the overall concrete strength and durability. When juxtaposed with traditional sand samples, the ITZ of the RS exhibited distinct characteristics, indicative of the unique microstructural nuances introduced by the recycled content, as shown in Figure 10. The morphology of the RS particles, characterized by varied sizes and shapes, was evident. While some irregularities were present, the general consistency of the particles pointed towards a uniform crushing process of the cubes. Notably, while slightly pronounced, the porosity around the RS particles remained within acceptable limits.

*3.11. Compressive Strength Test of Concrete (Prepared by the Mixing of Recycled Aggregate and RS)*

The compressive strength of concrete is a fundamental property often used to gauge the material's suitability for structural applications. It gives insight into the concrete's ability to resist axial loads and is a crucial metric in concrete structures' design and quality control. For the concrete prepared using recycled aggregate and RS, the compressive strength test results are as follows in Table 12.

**Table 12.** Quantitative Analysis of Axial Load Resistance from the Compressive Strength Assessment of Concrete.

| Sample No. | Weight (kg) | Density (kg/m$^3$) | Applied Load (kN) | Compressive Strength (N/mm$^2$) | Avg. Compressive Strength (N/mm$^2$) | Target Compressive Strength (N/mm$^2$) |
|---|---|---|---|---|---|---|
| 1 | 8.208 | 2432 | 600 | 26.67 | | |
| 2 | 8.192 | 2427 | 580 | 25.78 | 26.22 | 25 |
| 3 | 8.130 | 2409 | 590 | 26.22 | | |

The data show that all samples exhibited compressive strengths exceeding the target value of 25 N/mm$^2$. The average compressive strength across the three samples is 26.22 N/mm$^2$. This consistency in exceeding the target strength indicates that the concrete mix, derived from recycled aggregate and sand, is robust and maintains a high-quality standard. This finding underscores the potential of using recycled materials in concrete production without compromising the structural integrity of the resultant mix.

The findings of this research make a convincing case for sustainable construction practices, in line with SDG-9. Integrating RS and RCA in construction applications substantiates a pathway towards innovation and resilience in infrastructure. The AIV and strength of these materials under stress [58] meet the mechanical requirements and reverberate the need for sustainable industrialization [59]. This research contributes to the body of knowledge by demonstrating that the utilization of RS and RCA can participate in the race for sustainable infrastructure [60,61], pivotal to achieving the targets set by SDG-9 [62]. Aligned with the goals of SDG-11, this study also showed that RS and RCA can be considered critical components in sustainable urban development. The enhanced durability and stability demonstrated through the LAA Test [32] and the soundness test point towards their suitability for urban construction, addressing the need for long-lasting materials in dense, urban settings [63,64]. The successful application in high-traffic environments suggests a potential reduction in urban resource depletion, fulfilling the sustainable urbanization objective of SDG-11 [65]. Consistent with the principles of SDG-12, this research advocates for the sustainable consumption and production of construction materials [66]. The viability of recycled aggregates, as evidenced by their structural performance [67] and SEM analysis [68], encourages the construction industry to shift towards waste-reducing production methods [69]. The incorporation of Sinicon PP showcases how the industry has options to innovate towards more sustainable materials without compromising quality, thereby advancing toward the realization of SDG-12 [70,71].

## 4. Conclusions

The study affirms the potential of RCA and RS as sustainable substitutes for conventional building materials. By closely analyzing RCA and RS derived from repurposed concrete debris, we have highlighted their significant contributions to promoting greener construction methods without compromising material integrity or performance. The RCA's impressive AIV of 5.76% positions it as "exceptionally strong", making it suitable for high-impact applications. Similarly, the RS showcases a notable LAA value of 21.78%, evidencing its superior abrasion resistance. These findings support the argument that such recycled materials can replace traditional aggregates, contributing to environmental

sustainability and waste reduction in the construction sector. The study further explores the efficacy of RS when mixed with cement and Sinicon PP in a 3:1 ratio, uncovering an enhancement in fire and heat resistance. This property markedly increases the safety and longevity of construction materials. A comprehensive suite of tests—including evaluations of workability, impact, abrasion resistance, and compressive strength, alongside SEM for microstructural analysis—reveals the establishment of a robust bond between the cement paste and aggregates. This bond signifies the high quality of concrete achievable with RCA and RS, evidenced by a compressive strength of 26.22 N/mm$^2$ in M20-grade concrete. Soundness tests showing minimal expansion and a consistency level conducive to good workability further validate the practical utility of these recycled materials in construction mixes. This research proposes the onsite recycling of concrete waste into RCA and RS as a viable strategy for India's burgeoning construction industry. This practice not only addresses waste and emissions but also resonates with the global sustainable development goals, presenting an effective means to bolster environmental sustainability in construction. The alignment of the findings with SDGs 9, 11, and 12 underscores the role of RCA and RS in fostering sustainable construction practices. Their notable performance in terms of strength, durability, and environmental benefits advocates for a paradigm shift towards more sustainable construction methodologies. This transition not only supports the global sustainability agenda but also paves the way for resource-efficient and eco-friendly construction approaches. By suggesting onsite recycling of concrete materials, this research highlights the elimination of transportation costs and the promotion of local recycling initiatives. The remarkable properties of freshly recycled RS and RCA underscore their promise as sustainable construction materials. Thus, we advocate for their widespread adoption in construction projects, emphasizing the comprehensive benefits—environmental, economic, and qualitative—that these practices offer. We have proposed an important blueprint for the construction industry to contribute to the United Nations' SDGs, envisioning a future where construction builds infrastructure as well as steadfastly supports sustainability and environmental stewardship.

**Author Contributions:** Conceptualization, S.S. and S.K.S.; methodology, S.S.; software, S.A.M.; validation, G.M., S.K. (Shruti Kanga) and S.K. (Sujeet Kumar); formal analysis, M.M.; investigation, S.K. (Shruti Kanga); resources, P.K.; data curation, G.M.; writing—original draft preparation, S.S.; writing—review and editing, G.M.; visualization, S.A.M.; supervision, S.K. (Sujeet Kumar); project administration, P.K.; funding acquisition, G.M. All authors have read and agreed to the published version of the manuscript.

**Funding:** This research received no external funding.

**Data Availability Statement:** The data are available from the first author upon reasonable request.

**Acknowledgments:** We greatly appreciate the insightful suggestions provided by the three anonymous reviewers during the peer review process of this manuscript. Their input has significantly enhanced the quality of this work. Gowhar Meraj acknowledges the support of Japan Society for the Promotion of Science for JSPS KAKENHI (Grant Number 23KF0024).

**Conflicts of Interest:** The authors declare no conflicts of interest with Red Sea Global.

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
