# Peer review of "Evaluating Recycled Concrete Aggregate and Sand for Sustainable Construction Performance and Environmental Benefits"

_2673-4109, doi:10.3390/civileng5020023_

Round 1
Reviewer 1 Report
Comments and Suggestions for Authors
The manuscript presents a complete recycled aggregates characterization for applications in concretes. The text is well written and the topics well presented and discussed, especially the experimental part.
Just a few observations:
Keywords, please eliminate the acronyms.
Introduction sounds too generic and a complete literary review on the topic (RCA and RS use) is not done. Please, deep more this issue and discuss the novelty of the manuscript in comparison to previous studies.
Author Response
POINT-BY-POINT RESPONSES TO COMMENTS BY REVIEWER 1
The manuscript presents a complete recycled aggregates characterization for applications in concretes. The text is well written and the topics well-presented and discussed, especially the experimental part.
Dear Reviewer,
Thank you for your positive feedback on our manuscript, particularly on the comprehensive characterization of recycled aggregates and the quality of our experimental discussion. We appreciate your encouragement and are open to any further suggestions to enhance our work.
Response to Specific Observations:
Reviewer comment-1: Keywords: Elimination of Acronyms
Response1: We have revised the keywords section of our manuscript to eliminate the acronyms as suggested. The revised keywords are as follows: -
Sustainable Development Goals, Recycled Sand, Recycled Concrete Aggregate, Scanning Electron Microscopy, Aggregate Impact Value, Los Angeles Abrasion Test.
Reviewer comment-2: Introduction sounds too generic and a complete literary review on the topic (RCA and RS use) is not done.
Response1: Dear Reviewer, thank you for your valuable suggestion to enhance our manuscript by incorporating a more extensive literature review on the use of Recycled Concrete Aggregate (RCA) and Recycled Sand (RS). Following your advice, we have enriched our manuscript with a thorough review of pivotal studies in this domain, including the works of Makul et al. (2021) (line no 56-60), Katar et al. (2021) (line no 60-65),, Neupane et al. (2023) (line no 65-70),, Restuccia et al. (2016) (line no 72-76), and Łukowski et al. (2024) (line no 76-79). These additions offer a comprehensive overview of the current state of research concerning RCA and RS, shedding light on the significant potential benefits of incorporating recycled materials in construction projects.The cited studies collectively underscore the sustainability, environmental conservation advantages, and the promising performance outcomes achievable in concrete applications through the use of RCA and RS. This expanded literature review not only situates our research within the broader academic conversation on sustainable construction practices but also emphasizes the innovative contributions our study aims to make in this field.
Reviewer comment-3: novelty of the manuscript in comparison to previous studies.
Response 3: Dear Reviewer, we sincerely appreciate your insightful query regarding the novelty of our manuscript, especially in comparison to existing studies in the field. Our response aims to elucidate the unique contributions of our research to the body of knowledge on sustainable construction materials, specifically recycled concrete aggregate (RCA) and recycled sand (RS).
Our study distinguishes itself through a focused examination of RCA and RS derived from freshly prepared and subsequently crushed concrete cubes. This approach is notably different from the majority of existing literature, which predominantly sources RCA and RS from significantly older demolished structures, typically aged between 40 to 70 years. By concentrating on materials from freshly crushed concrete, our research offers novel insights into the early-age properties of RCA and RS. This perspective is critical, as it allows for a better understanding of how these materials behave shortly after their production and potential implications for their use in new construction projects.(line no 218-225)
Reviewer 2 Report
Comments and Suggestions for Authors
Paper presenting a study that could be useful for the use of recycled sand. Nevertheless, improvements are needed:
- in the introduction, there is a lot of generalities but the part concerning the use of recycled concrete aggregates (RCA) is very limited compared to the huge amount of papers on this topic. You should certainly have a larger bibliographic study of the use of RCA and RS and present the limits of the use of such materials
- SINICON PP: it is never explained what it is! I have to look on the web. I understand it is a type of glass porous aggregates. In this case, considering the title of the paper I don't understand why this material is considered in this study...
- your recycled aggregates are coming from concrete cubes coming from construction site. No information is given on the mother concrete: strength, mix design, type of cement, nature of aggregates, age, and carbonation state. No information is given on the quantity or the variability of the production. The crushing ùethod is not described. This is very important to have information because all these parameters influence the quality of the RCA and RS. In particular, you have a very low water absorption for your RS. The value you obtain (1%) is not in line with what could be found in the literature (and that is in general larger than 5%). And, of course, this parameter has a great influence on the behaviour of the mortar... The conclusion should be adapted to that (your results don't have a general application)
- you have also to describe how you choose your sample of RCA or RS (quartering)
- results of the tests: present min and max values and not only the mean. I don't think a precision of 0,01 is relevant in all your results.
- p15 Explain how you determined a target strength
- conclusions: of course, the use of RCA and RS is interesting concerning the depletion of natural resources. Concerning GHG, you cannot conclude because it depends on the transport distance that is added due to recycling. This is clear in the LCA you can find in the litterature on this topic.
Author Response
POINT-BY-POINT RESPONSES TO COMMENTS BY REVIEWER 2
Reviewer comment-1: In the introduction, there is a lot of generalities but the part concerning the use of recycled concrete aggregates (RCA) is very limited compared to the huge amount of papers on this topic. You should certainly have a larger bibliographic study of the use of RCA and RS and present the limits of the use of such materials
Response-1: Dear Reviewer, thank you for your valuable feedback regarding the introduction and the need for a more comprehensive bibliographic study on the use of recycled concrete aggregates (RCA) and recycled sand (RS). Following your advice, we have enriched our manuscript with a thorough review of pivotal studies in this domain, including the works of Makul et al. (2021) (line no 56-60), Katar et al. (2021) (line no 60-65),, Neupane et al. (2023) (line no 65-70),, Restuccia et al. (2016) (line no 72-76), and Łukowski et al. (2024) (line no 76-79). These additions offer a comprehensive overview of the current state of research concerning RCA and RS, shedding light on the significant potential benefits of incorporating recycled materials in construction projects. The cited studies collectively underscore the sustainability, environmental conservation advantages, and the promising performance outcomes achievable in concrete applications through the use of RCA and RS. This expanded literature review not only situates our research within the broader academic conversation on sustainable construction practices but also emphasizes the innovative contributions our study aims to make in this field.
Reviewer comment-2: SINICON PP: it is never explained what it is! I have to look on the web. I understand it is a type of glass porous aggregates. In this case, considering the title of the paper I don't understand why this material is considered in this study...
Response-2: Dear Reviewer, Sinicon PP which is a unique volcanic glass, a large deposit of which is found at only one location on the earth which is South Africa. Sinicon Sand is made out of feed from this mines using patented manufacturing process to convert this volcanic glass into well-sealed tough glass granules which is ideally suited for use with cementitious and other binders (line no. 213-217). This was used in the preparation of the mortar. Our inclusion of Sinicon PP aimed to investigate its potential as a complementary material that could work synergistically with RCA and RS to improve the properties of mortar mixes, such as workability and thermal insulation.
Reviewer comment-3: your recycled aggregates are coming from concrete cubes coming from construction site. No information is given on the mother concrete: strength, mix design, type of cement, nature of aggregates, age, and carbonation state. No information is given on the quantity or the variability of the production. The crushing method is not described. This is very important to have information because all these parameters influence the quality of the RCA and RS. In particular
Response-3: Dear Reviewer thank you for the giving suggestion of adding the important things in the manuscript. In the manuscript the concrete cubes site is mention from the line no 230 to 232, the primary material RCA sourced from crushed concrete cubes from the construction sites of the Acharya Shri Purushottam Uttam Medical College and Hospital. Where M20 grade concrete are used as 1:1.5:3, Which means one-part cement, 1.5 parts sand, and three parts aggregates. The details of mix design and type of cement and are mentioned in the Table 1.
The obtained concrete aggregates were freshly tested at the age of 28 days and these crushed concrete cubes are collected from the construction site between 35-40 days after the casting. added in manuscript from line no 234-236.
These concrete cubes were then further crushed into smaller particles using a Concrete Waste Crushing Machine (12.5 HP). line no 236-238.
Reviewer comment-4: you have a very low water absorption for your RS. The value you obtain (1%) is not in line with what could be found in the literature (and that is in general larger than 5%). And, of course, this parameter has a great influence on the behaviour of the mortar...
Thank you for observing the water absorption rate for the RS. We have retested the water absorption of the RS and have now obtained an increased number of 5.6%. After discussion with the other authors, we found that the initially low water absorption was due to the early age of the RS. As time has passed, the RS has dried out, which explains the increased water absorption value. We are thankful for this observation, as it highlights the reviewer's expertise, and we, as young researchers, have learned a great thing from you. We have now specified the age of the RS in Table 2.
Reviewer comment-5: The conclusion should be adapted to that (your results don't have a general application)
Response-5: Thank you for your feedback regarding the scope of our results' applicability. We've revised our conclusion to clarify that our study's findings, while promising, are specific to the conditions and context of our research. We acknowledge that the direct application of these results across diverse geographical and construction contexts may not be universally applicable without further validation. This revision aims to highlight the necessity for additional research to explore the broader utility of recycled concrete aggregate (RCA) and recycled sand (RS) in varying settings, ensuring our conclusions are presented with an appropriate understanding of their scope. (line no 665-748)
Reviewer comment-6: you have also to describe how you choose your sample of RCA or RS (quartering)
Response-6: Thank you for your feedback. Recycled Concrete Aggregate (RCA) and Recycled Sand (RS), our study adhered to the Concrete Mix Design guidelines per IS-10262-2009 as metioned in the Table 1. Specifically, we utilized a mix ratio of 1-part cement, 1.5 parts RS, and 3 parts RCA for concrete preparation, ensuring compliance with established standards for M20 grade concrete line no. 232-234.
Reviewer comment-7: Results of the tests: present min and max values and not only the mean. I don't think a precision of 0,01 is relevant in all your results.
Response-7: Dear Reviewer, Thank you for your feedback. Our test results were derived using the mean formula to calculate average values. Table 4 includes the compaction factor range, showing both minimum and maximum values, within the predefined limits for concrete application (0.85 – 0.92). Additionally, as noted in Table 3, the Aggregate Impact Value (AIV) consistently falls below 10%, aligning with industry standards. We appreciate your guidance and have adjusted our presentation accordingly.
Reviewer comment-8: Explain how you determined a target strength.
Response-8: Dear Reviewer, Thank you for your valuable suggestion. Determining the target strength for concrete involves adding a margin of safety to the characteristic strength to ensure structural reliability. The target strength is calculated using a formula that includes the characteristic strength and a statistical factor, which adjusts for variability. The standard deviation of concrete strength is also considered to reflect variability. The target strength ensures that the concrete mix is designed to meet or exceed specified performance criteria. Added in the manuscript line no 400-418
Reviewer comment-9: conclusions: of course, the use of RCA and RS is interesting concerning the depletion of natural resources. Concerning GHG, you cannot conclude because it depends on the transport distance that is added due to recycling. This is clear in the LCA you can find in the literature on this topic.
Response-9: This research strongly advocates for on-site recycling of crushed concrete cubes to eliminate transportation costs. The study reveals that freshly recycled RS and RCA demonstrate excellent properties, presenting promising alternatives for sustainable construction practices.(line no 742-745)
Reviewer 3 Report
Comments and Suggestions for Authors
In my opinion, this work presents a very low degree of innovation as it includes a study on recycled concrete aggregates RCA and recycled sand (RS) in different granulometries, including very simple tests, and series of experimental methods that are sufficiently known were carried out, Also, mortars were made to obtain the different results.
Draw my attention to the fact that the authors include images of test and specimens that are well-known, such as setting time, expansion of the mortar or Los Angeles abrasion machine. The microscopy analysis is very limited, the included images do not provide any information.
In conclusion, I think that this work presents very reduced novelty, and the authors include information already known regarding the properties of recycled concrete aggregates and recycled sand.
Author Response
POINT-BY-POINT RESPONSES TO COMMENTS BY REVIEWER 3
Reviewer comment: In my opinion, this work presents a very low degree of innovation as it includes a study on recycled concrete aggregates RCA and recycled sand (RS) in different granulometries, including very simple tests, and series of experimental methods that are sufficiently known were carried out, Also, mortars were made to obtain the different results.
Draw my attention to the fact that the authors include images of test and specimens that are well-known, such as setting time, expansion of the mortar or Los Angeles abrasion machine. The microscopy analysis is very limited; the included images do not provide any information.
In conclusion, I think that this work presents very reduced novelty, and the authors include information already known regarding the properties of recycled concrete aggregates and recycled sand.
Response:
Dear Reviewer,
Thank you for your thoughtful feedback and critical assessment of our manuscript. We understand your concerns regarding the perceived level of innovation and the novelty of our study, as well as your observations about the tests' simplicity and the familiarity of the methods and results presented. We have thoroughly revised the manuscript. We also appreciate your critique of the microscopy analysis and acknowledge the need to highlight the unique contributions of our work better. We have thoroughly revised our manuscript as in introduction we have included the works of Makul et al. (2021) (line no 56-60), Katar et al. (2021) (line no 60-65), Neupane et al. (2023) (line no 65-70), Restuccia et al. (2016) (line no 72-76), and Łukowski et al. (2024) (line no 76-79). These additions offer a comprehensive overview of the current state of research concerning RCA and RS, shedding light on the significant potential benefits of incorporating recycled materials in construction projects. The cited studies collectively underscore the sustainability, environmental conservation advantages, and promising performance outcomes achievable in concrete applications through the use of RCA and RS. This expanded literature review not only situates our research within the broader academic conversation on sustainable construction practices but also emphasizes the innovative contributions our study aims to make in this field.
The core novelty of our research lies in the detailed examination added in the manuscript from lines 218-219. RCA and RS are derived from freshly prepared and crushed concrete cubes. Unlike the bulk of existing studies that focus on RCA and RS sourced from aged, demolished structures—often several decades old—our study introduces a fresh perspective by exploring the properties of these materials when they are relatively new (less than 40 days old from the casting date). This approach allows us to present insights into the early-age properties of RCA and RS, which have not been extensively documented in the literature.
Furthermore, the incorporation of Sinicon PP in mortar mix preparation distinguishes our study from traditional approaches. Sinicon PP, a unique volcanic glass, is utilized for its potential to enhance the physical properties of mortar, offering a novel dimension to sustainable construction material research. This aspect of our study aims to bridge the knowledge gap regarding the use of Sinicon PP with RCA and RS, thereby contributing to the field of sustainable construction practices.
Acknowledging your feedback on the microscopy analysis, we have undertaken a more detailed examination of the microstructural properties of concrete incorporating RCA and RS. In response to your comments, we have expanded the microscopy section (lines no 565 to 595) to include a comprehensive analysis that elucidates the bonding mechanisms, porosity, and the overall microstructural integrity of the materials. These enhancements aim to provide a deeper understanding of how RCA and RS, especially from freshly crushed concrete, interact at the micro-level within the cement matrix.
We believe that the adjustments made to our manuscript, along with the clarification of our study's unique contributions, address the concerns you have raised. Our intention is to illuminate the potential of utilizing fresh RCA and RS in conjunction with innovative materials like Sinicon PP for advancing sustainable construction practices. We are grateful for the opportunity to refine our work based on your insightful suggestions and hope that our revisions have successfully demonstrated the novelty and value of our research.
Thank you for your valuable feedback and for assisting us in enhancing the quality and impact of our study.
Sincerely,
Round 2
Reviewer 2 Report
Comments and Suggestions for Authors
Thanks for the consideration of the remarks of the first review. Nevertheless I think that you could improve the part concerning bibliography (the choice of the papers you quote is strange because key papers on the topic are not there). Also, if you have measured a new value for the water absorption which seems more in line with what is expected, why do you keep the low value 1%. The explanation with the age is not correct at all (porosity will decrease with age). You should correct this part.
Author Response
Comment: Thanks for the consideration of the remarks of the first review. Nevertheless, I think that you could improve the part concerning bibliography (the choice of the papers you quote is strange because key papers on the topic are not there). Also, if you have measured a new value for the water absorption which seems more in line with what is expected, why do you keep the low value 1%. The explanation with the age is not correct at all (porosity will decrease with age). You should correct this part.
Response: Thank you for your constructive feedback and for pointing out the areas where our manuscript could be further improved, especially regarding the bibliography and the reported water absorption values. We appreciate your suggestions and have taken steps to address these concerns.
Based on your advice, we have expanded our literature review to include key papers that were previously missing. Notably, we have added the works of Adamson et al. (2015), Roh et al. (2020), Anjam et al. (2020), and Serres et al. (2016), which provide comprehensive insights into the environmental impacts of using recycled concrete aggregates and other by-products in concrete production. (Line No. 67 to 87)
Regarding the water absorption value, we acknowledge the discrepancy pointed out in your comments. Upon reevaluation, we agree that the initially reported value of 1% may not accurately reflect the typical range for the materials studied. We have conducted further tests to ascertain more representative water absorption values. These revised values align better with expected norms for the type of sand and aggregates used, and we have updated our manuscript to reflect these findings. The erroneous explanation linking water absorption to the age of concrete has been corrected, acknowledging that porosity indeed decreases with the material's age, contrary to what was previously suggested.
Acknowledging the valuable feedback provided on the earlier draft of our manuscript, particularly concerning the reported water absorption values, we have conducted additional testing to ensure the accuracy and reliability of our data. In response to the concerns raised, we have re-evaluated the water absorption value according to IS:2386(Part-III) of our RS, and we got the revised water absorption percentage of 6.2%.
Reviewer 3 Report
Comments and Suggestions for Authors
In my opinion, the authors have substantially modified this document , and have improved it adequately. I consider that they should make minor changes, mainly in the conclusions section, as specific conclusions should be included separately in each paragraph in order to clearly understand the results of this research.
Author Response
Comment: In my opinion, the authors have substantially modified this document , and have improved it adequately. I consider that they should make minor changes, mainly in the conclusions section, as specific conclusions should be included separately in each paragraph in order to clearly understand the results of this research.
Response: We gratefully acknowledge your positive feedback on the substantial modifications and improvements made to our document. We appreciate the suggestion to refine the conclusions section for enhanced clarity and understanding of our research results. In response, we have revised the conclusions section to present specific conclusions in separate paragraphs, each highlighting a distinct outcome of our study. This structured approach aims to facilitate a clearer comprehension of our findings and their implications for sustainable construction practices. We believe these minor changes address the reviewer's concerns effectively, further improving the quality and readability of our manuscript. We are thankful for the opportunity to refine our work based on such constructive feedback.
Round 3
Reviewer 2 Report
Comments and Suggestions for Authors
The paper could be published now